# The Role of TLRs in Obesity and Its Related Metabolic Disorders

**DOI:** 10.3390/ijms26052229

**Published:** 2025-03-01

**Authors:** Tannia Isabel Campos-Bayardo, Daniel Román-Rojas, Andrés García-Sánchez, Ernesto Germán Cardona-Muñoz, Daniela Itzel Sánchez-Lozano, Sylvia Totsuka-Sutto, Luis Francisco Gómez-Hermosillo, Jorge Casillas-Moreno, Jorge Andrade-Sierra, Leonardo Pazarín-Villaseñor, Wendy Campos-Pérez, Erika Martínez-López, Alejandra Guillermina Miranda-Díaz

**Affiliations:** 1Department of Physiology, University Center of Health Sciences, University of Guadalajara, Guadalajara 44360, Jalisco, Mexico; tanniaisabelcb@gmail.com (T.I.C.-B.); daniel.rrojas@academicos.udg.mx (D.R.-R.); andres_garciasanchez_3@hotmail.com (A.G.-S.); cameg1@gmail.com (E.G.C.-M.); itzel.10274@gmail.com (D.I.S.-L.); stotsuka@hotmail.com (S.T.-S.); 2Department of Laparoscopic Surgery, Hospital Civil de Guadalajara, “Juan I Menchaca”, Guadalajara 44360, Jalisco, Mexico; luisgomez_53@hotmail.com (L.F.G.-H.); jcasillas_moreno@hotmail.com (J.C.-M.); 3Department of Nephrology, National Medical Center of the West, Mexican Social Security Institute, Guadalajara 44340, Jalisco, Mexico; jorg_andrade@hotmail.com (J.A.-S.); leopazarin@hotmail.com (L.P.-V.); 4Department of Molecular Biology and Genomics, Institute of Nutrigenetics and Translational Nutrigenomics, University of Guadalajara, Guadalajara 44340, Jalisco, Mexico; wendy_yareni91@hotmail.com (W.C.-P.); erikamtz27@yahoo.com.mx (E.M.-L.)

**Keywords:** obesity, TLRs, metabolic syndrome, non-alcoholic fatty liver disease, atherosclerotic disease, chronic inflammation

## Abstract

Obesity affects the adaptability of adipose tissue (AT), impairing its ability to regulate energy and metabolism. Obesity is associated with many metabolic disorders, including dyslipidemia, hypertension, sleep disorders, non-alcoholic liver disease, and some types of cancer. Toll-like receptors (TLRs) are important in obesity and related metabolic disorders. TLRs are pattern-recognizing receptors (PRRs) involved in the innate immune system and recognize pathogen-associated molecular patterns (PAMPs) and endogenous ligands. TLRs, especially TLR2 and TLR4, are activated by fatty acids, endotoxins, and other ligands. TLR2 and TLR4 activation triggers inflammatory responses. Chronic inflammation driven by TLR activation is a hallmark of obesity and metabolic diseases. The inflammatory response triggered by TLR activation alters insulin signaling, contributing to insulin resistance, a key feature of metabolic syndrome and type 2 diabetes. Modulation of TLR activity through lifestyle changes (diet and exercise), obesity surgery, and pharmacological agents is under study as a possible therapeutic approach to controlling obesity and its complications.

## 1. Introduction

The World Health Organization (WHO) has reported obesity as a global health problem since 2018 [1]. The accelerated modernization of countries favors a better standard of living. However, modernization often comes hand in hand with bad health habits, including bad eating habits [1]. Obesity is considered a more prevalent non-communicable disease [2]. In just a few decades, obesity has become a pandemic that threatens multiple aspects of people’s health from different demographics. Obesity affects people regardless of age and contributes exponentially to the worldwide prevalence of type 2 diabetes mellitus (DM2) [3].

Obesity provides the metabolic conditions for the development of comorbidities such as cardiovascular disease, insulin resistance, diabetes, non-alcoholic fatty liver disease (NAFLD), and cancer [4]. Obesity results from an energy imbalance between caloric intake and caloric expenditure. In addition, this energy imbalance is accompanied by other metabolic alterations such as oxidative stress, mitochondrial dysfunction, immune dysfunction, and chronic low-grade inflammation [5]. The excess calories are stored in adipose tissue (AT). This AT responds to over-nutrition through inflammatory responses that trigger inflammatory processes through mechanisms that are not fully understood [6]. White adipose tissue (WAT) is the main fat storage reservoir and has endocrine functions by secreting adipokines and cytokines. The function of adipokines, together with insulin, is metabolic and physiological signaling of glucose uptake, fatty acid oxidation, and other energy-yielding metabolic processes [7]. Cytokines regulate inflammation and adaptive and reparative angiogenesis. Excess weight also modifies the phenotype of WAT, causing inflamed and dysfunctional adipocytes [8]. Inflamed adipocytes secrete proinflammatory cytokines that modify the normal homeostasis of the AT and other organs systemically [9]. Excessive caloric intake causes increased body fat and inflammation of the AT in humans and animals [10]. This paper will briefly discuss the role of (TLRs) in obesity. Initially, we will address the topic of inflammation in obesity, describe the role of each TLR in obesity, TLRs in atherosclerotic vascular disease, TLRs in obesity-related insulin resistance, TLRs in obesity-associated T2DM, TLRs in obesity-related NAFLD, and TLRs in cancer and obesity.

## 2. Search Strategy

The search for information was carried out using mainly the following electronic databases: PubMed, Scopus, and Google Scholar. The following keywords were used in the search: TLR, TLR1-10, obesity, adipose tissue, metabolic syndrome, metabolic disease, atherosclerosis, endothelial dysfunction, diabetes, insulin resistance, NAFLD, NASH, fibrosis, inflammation, oxidative stress, treatment, therapeutic, clinical trial, disease, mechanism, vascular disease, cancer. The AND/OR connection operators were used between the keywords. The keywords were searched with the “Title” and “Abstract” filters. The search for information was complemented by the review of the reference lists of the articles found. Aggregate information on general concepts was also searched in parallel. Articles on clinical trials and in vivo and in vitro experimental studies were included. Articles in languages other than English and Spanish were excluded. No performance indicators related to the research topic were carried out.

## 3. Inflammation in Obesity

To approach the subject from a general perspective, we will initially mention the inflammation process in obesity. Inflammation is known for its role in cellular and molecular responses that defend the body against infections and other threats. These inflammatory mechanisms also contribute to the repair of damaged tissues [11]. Immune cells can have proinflammatory or anti-inflammatory activity, and both functions control inflammation processes [12]. Inflammation is a process where energy expenditure increases and energy intake is reduced directly and indirectly. Inflammation is a process in which energy expenditure increases and energy intake is decreased directly and indirectly. Inflammatory cytokines such as tumor necrosis factor-alpha (TNFα), interleukin-1 (IL-1), and interleukin-6 (IL-6) signal an increase in energy expenditure in the central nervous system (CNS) and other organs with high metabolic activity [13]. Leptin expression increases in AT due to inflammatory cytokine activity [14]. TNFα mediates the increase in leptin receptor expression, which increases leptin activity to promote energy expenditure in different tissues [15]. Leptin induces energy expenditure indirectly by acting on the CNS to decrease appetite [16]. However, the unique characteristics of over-nutrition-induced AT inflammation are not due to increased energy expenditure, making it possible for weight gain and inflammation to occur mutually [17]. Inflammation of WAT causes infiltration of immune cells from the bone marrow and secretion of inflammatory agents (chemokines and cytokines) from adipocytes and resident immune cells. The release of inflammatory cytokines can cause systemic inflammation even in non-obese subjects [18]. The inflammatory phenotypic properties of WAT are more intense in visceral white adipose tissue (VWAT) than in subcutaneous white adipose tissue (SWAT). A phenotypic characteristic of VWAT is the higher number of macrophages compared to SWAT [19]. Furthermore, increased body mass and insulin resistance lead to more severe hypertrophy in VWAT adipocytes than in SWAT adipocytes [20]. In obese subjects, the phenotype of VWAT adipocytes tends to be predominantly inflammatory, which may reduce lipogenic markers and lead to metabolic complications such as ectopic lipid deposition in skeletal muscle and liver [21]. Increased ectopic deposition also adds problems in peripheral insulin signaling and complications related to systemic insulin resistance and the development of type 2 diabetes [20].

Human studies have described the heterogeneity between VWAT and SWAT-free fatty acid (FFA) absorption metabolism. FFA absorption from meals is more prominent in VWAT than in SWAT in both men and women [22]. In the adrenocortical region of women, direct fat absorption is more notable in the omental structure than in the subcutaneous portion. The convenient anatomical location of VWAT allows different abdominal organs, such as the liver, to directly receive proinflammatory cytokines and other metabolites to regulate systemic metabolism [23]. For these reasons, obesity is considered a state of low-grade chronic inflammation that favors an imbalance of different metabolic processes and the appearance of diseases that alter the quality of life and increase healthcare costs [24,25]. One of the frequent metabolic alterations in obesity is insulin resistance [26].

Insulin resistance is characterized by the inability of various tissues to respond to normal circulating insulin levels adequately. The first observed abnormality of insulin resistance is the insulin-induced decrease in glucose uptake in AT and skeletal muscle [27]. Impaired insulin sensitivity has multi-organ effects. AT is one of the main tissues that develop hypertrophy, causing increased serum and tissue levels of adipokines and proinflammatory molecules such as IL-6 and TNF-α [28]. Studies show that overexpression of TNF-α induces insulin resistance and improves when it is neutralized [29].

In obese subjects, a decrease in anti-inflammatory molecules such as adiponectin can be found; this decreased expression is associated with cardiovascular disease (CVD) [30]. Mainly, FFAs are transported free (unesterified) or bound to other carrier molecules (lipoproteins). These FFA increase significantly in obesity and are responsible for increasing AT in response to an alteration of the antilipolytic effect of insulin and less re-esterification of FFA by adipocytes in obesity [31]. FFAs are found in circulation as saturated fatty acids (SFA), monounsaturated fatty acids (MUFA), and polyunsaturated fatty acids (PUFA), but their proportions vary. The chain length of FFAs can be short, medium, and long. However, the amount of carbon in FFAs is closely related to its effects on immune immunity [32]. The immune system recognizes PAMPs from viruses through PRRs, including transmembrane proteins called Toll-like receptors (TLRs). These TLRs are located both in the cell membrane (TLRs 1, 2, 4, 5, 6) and intracellular endosomes (TLRs 3, 7, 8, and 9) [33]. obesity, an association has been demonstrated between the concentration of cytokines IL-6 and TNF and the expression of TLRs [34].

## 4. TLR

TLRs have a similar structure to Toll receptors discovered in the fruit fly Drosophila melanogaster [35]. TLRs are composed of three structural domains composed of glycoproteins. One of the domains is a leucine-rich repeat (LRR) domain, another is a transmembrane helical domain, and the third is the Toll-interleukin receptor (TIR) domain. The LRR domain is the binding site for intruding molecules, while the TIR performs signaling functions [36]. PRRs recognize different PAMPs and enable TLRs to support innate immune responses by detecting different signals from exogenous microorganisms [37,38]. TLR-mediated signaling pathways also modulate different cellular activities, including glucose and lipid metabolism [39]. Adipocytes express TLRs, and immune system cells are positioned between obesity and inflammation [40]. PRRs allow TLRs to recognize several different ligands [41]. Humans possess 10 members of the TLR family. TLRs 1, 2, 4, 5, and 6 are located in the plasma membrane and primarily detect microbial molecules and PAMPs that stimulate the expression of proinflammatory molecules. TLRs 3, 7, 8, and 9 are located in endosomal compartments and detect viral and bacterial nucleic acids and respond with the production of type 1 interferon (IFN) [42]. Activation of infiltrating immune cells leads to the relocalization of white blood cells and macrophages to manage inflammatory processes and the activation of mature adipocytes [43]. Different TLRs recognize specific ligands. However, they share many of the cellular signaling pathways that they mediate. All TLRs activate signaling by adaptor molecules called myeloid differentiation 88 (MyD88) and Trif, which elicit similar cellular responses to different ligands. Many TLRs serve primarily to enable PAMPs and not to generate personalized immune responses [44].

TLRs are involved in different infectious and non-infectious inflammatory processes related to obesity and immune responses to AT [45]. Low-grade inflammation in obesity may be linked to an activation of circulating TLRs and ATs caused by microbial pathogens that modify intestinal structure and permeability [46]. Depending on the expression of each type of immune cell and adipocyte TLR, changes in anabolic and catabolic processes can be generated to favor energy production for the immune system [47]. TLR effects produce cytokine and chemokine production signaling responses that initiate inflammatory and immunological actions against infections [37]. Cellular mechanisms mediated by TLRs ultimately activate signaling cascades that target transcription factors of proinflammatory molecules [48]. The molecules activated by TLRs include nuclear factor kappa B (NFκB), IL-1R-associated kinase (IRAK)-1, IRAK-4, IκB complexes, and tumor necrosis factor-associated actor (TRAF)-6 [49]. Each type of TLR expression in visceral AT specifically involves inflammatory and metabolic problems in obesity. Evidence associates the increased expression of TLRs, except TLR3, with inflammation and decreased insulin sensitivity in obesity-induced AT [45]. TLRs contribute to the organism’s immune defense by detecting microorganisms’ presence. For this reason, the effects of TLR expression in adipocytes on the complex mechanisms of inflammation and obesity-associated diseases have been studied [21]. It is known that TLRs induce the expression of proinflammatory molecules such as cytokines and that omental tissue is where inflammation manifests to a greater degree. In this sense, the expression of TLRs in omental tissue may be closely linked to the mediation of this localized inflammation [21]. TLR4, TLR1, TLR2, and TLR6 are significantly overexpressed in omental adipose tissue. Previously, the preferential infiltration of macrophages in omental fat versus obese adipose tissue was demonstrated [19]. The hypothesized mechanisms suggest that some exogenous lipopolysaccharides (LPS) may interact in omental tissue, functioning as ligands for TLR overexpression in human omental tissue, but other TLR ligands could also be involved [50]. Endotoxemia has been shown to be actively involved in the onset of obesity and insulin resistance [45]. TLR2 and TLR6 during the differentiation of fibroblast cells called 3T3-L1 into adipocyte-like cells [51]. On the other hand, in fully differentiated adipocyte cells, the expression of TLR1 and TLR4 predominates [52]. In obesity, inflammation, and cardiometabolic dysfunction are related to the same TLR-mediated signaling pathways [53]. Many molecules orchestrate the energetic metabolic processes in obesity, and insulin is one of the most prominent. Insulin, in addition to its control over glucose, indirectly helps to decrease NFkB activity, reducing reactive oxygen species (ROS), p47phox, and C-reactive protein (CRP), which is interpreted as anti-inflammatory activity [54]. At some point, these mechanisms can be linked to the effects of some TLRs. The general metabolic effects of each TLR are described below Figure 1.

### 4.1. TLR1

The TLR1 gene shares characteristics with the TLR6 gene, and its PRR recognition activity is carried out by forming a heterodimerization complex with TLR2. TLR1 is found in the cell membrane and mediates innate immune responses to bacterial lipopeptides. The TLR1-TLR2 complex signals the activation of proinflammatory molecules through the activation of NFκB and the secretion of cytokines [59].

### 4.2. TLR2

TLR2 can recognize tri- and diacylated lipopeptides PAMPs by forming heterodimers with TLR1 or TLR6 [60]. TLR2/1 heterodimer formation recognizes triacylated lipopeptides (LPs) from the Gram-positive bacterial cell wall, while TLR2/6 heterodimer identifies diacylated ones. Both TLR2/1 and TLR2/6 increase the expression of proinflammatory cells by MyD88-dependent signaling [61,62]. Bacterial lipoproteins, lipomannans, and lipoteichoic acids are recognized to a greater extent by TLR2 [14]. Some metabolic complications, such as insulin resistance and β-cell dysfunction, are related to the overexpression of TLR2 and TLR4 in patients with obesity and DM [15,16,63]. MyD88-dependent signaling is initiated by TLR2 activation via SFA, both for the TLR2/1 and TLR2/6 heterodimers. This stimulation favors the expression of inflammatory cytokines such as IL-6 [64].

Evidence shows that TLR2 and NF-κB can activate the expression of proinflammatory molecules in human differentiated adipocytes [65]. Furthermore, studies in human atherosclerotic plaques show that TLR2 and TLR4 are predominantly expressed [66]. In mouse studies, atherosclerotic lesions and DM are associated with TLR2 and its genetic variants [58]. TLR participates in the immune response by being able to recognize gram-positive bacteria by their lipopeptides, peptidoglycans, and other lipids [67]. However, in experimental animal studies, TLR expression has been associated with myocardial tissue injury due to ischemia-reperfusion [68].

TLR2 depletion is associated with reduced experimental myocardial infarct size and prevention of abnormal ventricular remodeling [69]. In patients with T2DM, the increased expression of TLR2, TLR4, and plasma LPS is associated with plasma insulin concentration and insulin resistance [70]. Nutritional fatty acids can stimulate TLR4 and TLR4 [71]. In vitro and in vivo experiments show that TLR2 and TLR4 have opposing regulation in signaling adipocyte differentiation [72]. However, the relevance of the effect of TLR2 and TLR4 on conditions such as insulin resistance is not entirely clear, with some studies indicating that TLR2 is a protective factor [57] and others indicating the opposite [73]. In any case, the participation of TLR2 in the differentiation of 3T3-L1 fibroblast cells towards adipocyte cells is present [74].

TLR2 recognizes many ligands from specific bacterial strains, including lipoteichoic acids and various proteins, including lipoproteins and glycoproteins, zymosan, peptidoglycan, and LPS [75].

### 4.3. TLR3

TLR3 is a type of endosomal TLR with specificity for double-stranded RNA of viruses and is highly expressed in pancreatic β-cells. Metabolically, TLR3 is related to inflammation processes and impaired glucose tolerance [76]. Decreased TLR3 expression in WAT was recently reported in obese individuals with and without T2DM compared to lean individuals [77]. This decrease in TLR3 expression and decreased leptin receptors are also shown in human AT in a setting of obesity-related inflammation [78]. Therefore, the decrease in TLR3 in AT results in poor detection of double-stranded RNA of pathogens and altered immunoregulatory capacity of adipocytes [78]. TLR3 appears to regulate the AT homeostasis of the inflammatory state, but this regulatory property is altered in obesity [79]. However, downregulation of TLR3 may have a protective effect in atherosclerotic disease due to decreased macrophage mediation and improved collagen concentration and repair in atherosclerotic plaque lesions [55].

### 4.4. TLR4

TLR4 was the first human TLR homolog cloned and characterized in 1997. It was recently shown to be a receptor for recognizing cell wall components of gram-negative bacteria. TLR4 can sense FFA and induce insulin resistance in AT [80]. LPS predominantly activates TLR4. TLR4 and elevated FFA may be linked to NFκB-dependent pathogen-sensing activation, triggering inflammatory processes in non-pathological conditions [56]. TLR4 deficiency has protective effects on insulin resistance in mice, but TLR4 polymorphisms in humans increase the risk of T2DM [81]. TLR4 can bind to other molecules, such as resistin, a cytokine that has shown a role in the physiology of obesity and DM [82]. TLR4 polymorphisms are also associated with other health conditions, such as atherosclerosis and cardiovascular events [83]. TLR4 function in human AT is limited, but its expression is increased in atherosclerotic plaques, including monocytes, vascular cells, and platelets [84]. TLR4 polymorphisms result in decreased receptor signaling and decreased circulating proinflammatory mediators, which protect against carotid atherosclerosis but increase susceptibility to bacterial infection [85]. TLR4 also recognizes endogenous oxidized fibrinogen and low-density lipoprotein, which may promote atherothrombotic events. Atherosclerotic lesions express high concentrations of TLR4. Increased TLR4 expression is also observed in human lipid-rich coronary macrophage-infiltrated plaques [85]. In murine models, high levels of fatty acids can activate TLR4 signaling in fat cells and macrophages, resulting in insulin resistance [86]. TLR4 variants show a role in cardiovascular disease because they are associated with an increased risk of carotid intimal thickening and myocardial infarction [85]. TLR4 is also expressed in non-immunological cells, such as cardiac myocytes, which are positively regulated in hearts with heart failure [87]. In the metabolic obesity of mice, failure of TLR4 protects against insulin resistance and diet-induced obesity (DIO) [88]. The overload-induced inflammation and metabolic disease processes are related to signaling by the binding of TLR4 to SFE and hormone resistin [89]. Fatty acids participate in the induction of TLR4 expression during adipocyte differentiation [90]. TLR4 recognizes LPSs, which conform to much of the cell wall of gram-negative bacteria. Both TLR4 and LPS activate the mediators MyD88 and TIR domain-containing IFN-β-inducing adaptor (TRIF), which provides activation of signaling pathways [91]. The MyD88-dependent signaling pathway activates proinflammatory transcription factors such as NF-κB and AP-1, which leads to inflammatory processes mediated by cytokines and chemokines [92]. SFA-induced inflammation can be attenuated by impairment of TLR4 [93]. In addition to TLR4, the heterodimeric activation of TLR2/TLR1 can also impede the differentiation of fibroblast cells into adipocytes [94].

TLR4 and TLR2 are activated by SFA [6]. TLR4 and TLR2 are involved in bacterial lipoprotein recognition [95]. Activation of TLR4 and TLR2 by LPS shows a cross-regulation in 3T3-L1 fibroblast cells [96]. TLR4 can specifically recognize the lipid A portion of LPS of some viruses, such as mouse mammary tumors, vesicular stomatitis, and respiratory syncytial [97]. Impaired intestinal permeability due to changes in microbiota may increase the circulation of TLR agonist endotoxins in the blood [98].

### 4.5. TLR5

During endotoxemia, TLR5 recognizes bacterial flagellin and, together with TLR2, TLR4, and TLR2, mediates immune responses to other exogenous lipoproteins [99]. Evidence shows that TLR5 is present in high concentrations in atherosclerotic lesions [100]. Fat-induced atherosclerosis can be experimentally ameliorated when TLR5 deficiency is induced in a mouse model [101].

### 4.6. TLR6

TLR6 identifies specifically gram-positive bacteria structures. TLR6 specifically recognizes components of gram-positive bacteria. Despite the possible implications of TLR2 in the pathogenesis of NAFLD, few have studied the role of TLR6 [102]. Dysregulated peripheral TLR6 expression and activity in morbidly obese patients may reflect the well-known hepatic inflammatory event drivers of obesity-related NAFLD pathogenesis. NAFLD patients present high expression of TLR6 in hepatocytes compared to normal subjects. Dysregulation of TLR6 levels may be associated with TLR2-mediated liver inflammation in NAFLD. Future investigation of the effects of TLR6 on NAFLD may point to TLR6 as a biomarker for obesity-related NAFLD [102]. TLR6 is also significantly overexpressed in hepatocytes from NAFLD patients compared to normal subjects. Dysregulated TLR6 expression may potentiate TLR2-mediated hepatic inflammation in the pathogenesis of NAFLD. TLR6 could be a potential peripheral biomarker of obesity-related NAFLD [102]. In the immunological aspect, TLR6 is associated with the leading single nucleotide polymorphisms (SNPs) related to symptoms of allergic sensitization [103,104].

### 4.7. TLR7 y TLR8

TLR7 and TLR8 are responsible for recognizing short single-stranded and double-stranded RNA of viruses that are trapped within the endosomal chamber. The main activity of TLR7 and TLR8 is to mediate the expression of proinflammatory moieties of monocytes and lymphocytes. In AT, TLR8 expression is elevated in subjects with obesity or diabetes, increasing inflammatory processes [105].

### 4.8. TLR9

TLR9 is found in the membrane of endosomes and is responsible for identifying microbial DNA molecules, primarily the unmethylated CpG DNA structure [106]. Recognition of bacterial DNA and chromatin allows TLR9 to mediate adaptive immune and autoimmune responses [107]. Autoimmune mechanisms in pancreatic islets have been linked to TLR9 expression in experimental type 1 DM [108].

### 4.9. TLR10

Unlike previously reviewed TLRs, TLR10 has no confirmed ligands or biological function. However, TLR10 expression is stimulated by hydrogen peroxide-induced oxidative stress in monocytes and peripheral blood mononuclear cells. A positive correlation has also been found between TLR4 expression and body mass index (BMI) in patients with obesity or T2DM [109]. Some studies have found a relationship between TLR10 polymorphisms and inflammatory diseases such as asthma and Crohn’s disease [110,111].

### 4.10. TLR in Atherosclerotic Vascular Disease

Atherosclerotic vascular disease (ASVD) is a disease related to the long-term progression of endothelial damage that, over time, can trigger important complications such as myocardial infarction, acute coronary syndromes (ACS), and stroke [112]. Endothelial cell (EC) damage can be generated by hypercholesterolemia, hypertension, and vascular bifurcation disorders. The increase in lipids such as cholesterol interacts with the walls of damaged vessels and generates the appropriate environment for local inflammation [113]. In ECs of damaged vessels, signaling for leukocyte adhesion mediated by P-selectin and vascular cell adhesion molecule-1 (VCAM-1) is initiated [114]. Chemotaxis of monocytes activated by proinflammatory molecules guides the path to the vascular wall for subsequent differentiation into macrophages. In the area of the damaged vessel, phagocytosis of modified lipoproteins (oxidized low-density lipoprotein (ox-LDL)) leads to the formation of foam cells [115]. Leukocytes and vascular ECs release fibroblast growth regulatory factor (FGF), which activates the formation of vascular smooth muscle cells (VSMC), which migrate towards the inner elastic layer to the sub-intima of arteries. VSMC then expresses adhesion factors and cytokines [116]. Inflammatory cytokines and matrix metalloproteinases (MMPs) present in the advanced stage of atherosclerosis can degrade extracellular matrix (ECM) proteins and rupture the plaque. The formation of blood vessels in the atheromatous plaque is mediated by the secretion of vascular growth factor (VGF) and can lead to complications such as myocardial infarction or stroke [117].

Research literature points to chronic infections being closely linked to the development of atherosclerosis [118]. There is evidence that pathogenic viruses or bacteria are present in atherosclerotic plaques. These pathogens are hypothesized to alter the proper cellular environment and damage the local plaque region [119]. Viruses and bacteria activate signals for the formation of macrophage-derived foam cells by inhibiting apoptosis by inflammatory responses of ECs. However, other factors contribute to atherosclerotic plaque formation by microbial infection, such as genetics or susceptibility to vascular damage and diet. The origin of atherosclerotic plaques remains multifactorial [119].

Although a relationship between atherosclerotic plaque formation and pathogenic bacteria has been demonstrated, studies with anti-infective therapy show no benefit against atherosclerotic cardiovascular events. Therefore, understanding the TLR signaling mechanisms takes on a more significant interest in recognizing pathogenic bacteria and forming atherosclerotic plaques [120]. TLRs are PRRs at the plasma membrane and in the cytoplasm and participate in the innate immune response [121]. PRRs are specialized in recognizing PAMPs from bacterial LPS, RNA from viruses, or molecules generated by cellular damage or necrosis called danger-associated molecular patterns (DAMPs) [122].

TLRs are relevant in atherosclerosis because they can mediate specific immune responses by recognizing specific components of bacteria and viruses. This recognition of exogenous components tightly orchestrates inflammatory processes and immune defense against pathogens. Inappropriate TLR activity may contribute to the formation of atherosclerosis [123].

Atherosclerosis is associated with the down-regulation of TLR4 in ApoE^−/−^ and LDLR^−/−^ mice, while TLR2-TLR1 and TLR2-TLR1 heterodimers are shown to promote it [124]. TLR9 propagates inflammation when appropriate conditions of foam cell accumulation are generated, accompanied by vascular injury in LDLR^−/−^ mice [125]. Some studies show the protective effect of endoplasmic TLRs in atherosclerosis processes. For example, TLR3 seems to have a protective effect in the formation of atherosclerosis induced by mechanical and hypercholesterolemia-arterial injury [126]. TLR7 is associated with decreased macrophage-mediated inflammatory activity and cytokine production [127]. TLR9 activation decreases the severity of atherosclerosis in ApoE^−/−^ mice fed with a high-fat diet (HFD). TLRs induce changes in different mechanisms of atherogenesis that can lead to foam cell formation, vascular cell dysfunction, reduction or increase in inflammatory processes, recruitment of macrophages, and plaque instability, ultimately resulting in protective or aggravating effects on atherosclerosis [128].

### 4.11. TLR in Insulin Resistance in Obesity

Insulin resistance is characterized by the body’s failure to respond to the physiological activity of insulin [129]. Insulin resistance is highly prevalent in individuals with obesity, hypertension, and CVD. Insulin resistance chronically escalates in the development of T2DM. Insulin resistance is a disturbance of cellular metabolism that manifests in cells of different tissues, including the AT, muscle, and liver [129]. The liver is the master controller of gluconeogenesis and glycogenolysis to maintain healthy fasting glucose levels. An insulin-resistant liver has impaired control of gluconeogenesis and glycogenolysis, leading to elevated glucose concentrations. Similarly, when insulin resistance is present, the AT and muscle tissue cannot access blood glucose for normal cellular processes, leading to metabolic disturbances in those tissues [130]. In insulin resistance, pancreatic β-cells attempt to overcome the diminished effects of insulin by producing more of the hormone. However, β-cells have a limit of insulin production that cannot be exceeded. This overproduction of insulin can develop into T2DM, where insulin production is very limited compared to the demand for glucose in the blood [130].

Insulin-sensitive cells from different organs and tissues trigger an impairment of the insulin signaling pathway when they develop insulin resistance. In body cells, insulin binds to its membrane receptor, which is activated by autophosphorylation at the Tyr residues of insulin receptor substrates (IRS) [131]. IRS drives a linear signaling cascade that activates Akt by mediating PI-3 kinase or SHP2. Akt activation induces glycogen synthesis through Glut4 translocation, leading to metabolic signaling pathways. Insulin resistance and T2DM result from the disruption of insulin signaling [131]. Although insulin resistance is associated with increased body mass, the exact mechanisms of the origin of insulin resistance caused by obesity, OS, and mitochondrial dysfunction are still unknown [132]. Some mechanisms, such as dysregulation of lipid homeostasis, cellular hypoxia, and endoplasmic reticulum stress, have been investigated as mechanisms of origin of insulin resistance. Inflammation-induced by obesity may also be involved in the mechanisms that produce insulin resistance [132].

Increased FFA in AT, low HDL cholesterol, and VLDL hypertriglyceridemia have been associated with insulin resistance. This increase in FFA may promote endothelial dysfunction along with decreased L-arginine and/or NO and OS [133]. Increased levels of most polyunsaturated FFAs lead to endothelial cells undergoing cell cycle arrest and apoptosis. Insulin resistance disrupts insulin PI-3 kinase and Akt-dependent signaling pathways. Furthermore, hyperinsulinemia increases mitogen-activated protein kinase (MAPK) pathway activity. These changes in insulin signaling in endothelial cells decrease NO production and increase ET-1 concentration. Hyperinsulinemia promotes ET-1 secretion through MAP kinase activation and increases E-selectin and VCAM-1 [134]. In obesity, low-grade systemic inflammation is established together with dysregulated insulin signaling. TLR2/TLR4 binding with increased SFA in hyperlipidemia may trigger inflammatory processes [43]. SFA interaction with TLRs favors the mammalian target of rapamycin (mTOR) activation, macrophage metabolic reprogramming, and lipid metabolism, contributing to macrophage training. Increased intestinal permeability may occur in obesity and cause elevated circulating endotoxins, supporting macrophage training [135]. FFA accumulation by incomplete metabolism of fatty acid intermediates can activate different proinflammatory signaling pathways such as IκB β-kinase (IKKβ), protein kinase C (PKC), and Janus kinase (JNK) as well as decrease insulin metabolism signaling by IRS-1 in vivo cultures [87]. mTORC1 signaling mediates inhibition of the PI3K/AKT pathway and is activated by increased amino acids, insulin, and pro-inflammatory cytokines caused by overnutrition in obesity [136]. The transcription factor CCAAT enhancer binding protein b (CEBPb) controls the expression of many anti-inflammatory genes, including the TLR regulator IRAK3. Different metabolic pathways may participate to some extent in the control of TLR responses [137].

### 4.12. TLR in Diabetes and Obesity

T2DM is a worldwide disease capable of disabling and leading to premature death [138]. T2DM is considered a redox and inflammatory disease [139]. Research to understand the relationship between obesity and DM points to the innate immune system response activated by excess nutrients in most organs that regulate the body’s energy [140]. VWAT is a major source of proinflammatory molecules in DM. Hyperglycemia triggers metabolic processes such as macrophage infiltration and hypoxia, which increase cytokines and chemokines [141]. The increase in inflammatory factors in DM impairs the secretory function of pancreatic β-cells, which decreases the amount and effects of insulin [142]. Chronic systemic obesity inflammation can lead to β-cell dysfunction, insulin resistance, and T2DM. As previously mentioned, insulin resistance is the precursor to T2DM. Progressive insulin decline is a strong marker of pancreatic β-cell dysfunction, leading to impaired glucose metabolism and T2DM. T2DM is primarily a result of the effects of insulin resistance and pancreatic β-cell dysfunction. Inflammation in obesity is closely related to comorbidities of DM, such as retinopathy, nephropathy, NAFLD, and cardiovascular disease [133].

T2DM has also been linked to some autoimmune diseases, such as rheumatoid arthritis, Alzheimer’s disease, and polycystic ovary syndrome. Glucose control is the first target to prevent complications of T2DM, but it is not the only target [129]. Chronic inflammation is a systemic condition observed in tissues such as pancreatic islets, liver, muscle, and AT that causes metabolic and immunological alterations. The literature has named “immunometabolism” the interrelationship between metabolic and immune processes [26]. The accumulation of macrophage cells in energy-metabolizing tissues increases the release of proinflammatory cytokines and generates a state of chronic low-grade inflammation [26]. In the presence of obesity, inflammation driven by macrophages and other immune cells occurs in the AT, liver, and muscles. The pro-inflammatory cytokines released by macrophages function as autocrine and paracrine molecules that decrease insulin signaling from nearby tissues and promote β-cell dysfunction. Increased interleukin-1β is associated with T2D complications [143].

Endothelial vascular disease in DM is a disease that arises from the effects of endothelial dysfunction [144]. The function of the endothelium is to delimit the circulating blood from the lumen and the vascular wall through a monolayer of EC. This monolayer of blood vessels provides a homeostasis regulation of inflammatory activity, cell proliferation, fibrinolysis, balancing coagulation, and vascular permeability. Vascular smooth muscle cells and cells found in the bloodstream (macrophages, erythrocytes, leukocytes, renal mesangial cells, and retinal pericytes) also respond to signaling by chemical mediators of EC. When the endothelium is damaged, the body responds with EC repair mechanisms [145]. However, in DM, this endothelial repair process is altered in addition to the appearance of microvascular cells [146]. A deficiency in the repair process impairs the adhesion of blood cells to ECs, increases the sensitivity to apoptosis of ECs, and violates the barrier function of the endothelium [147]. The consequences of damage to the properties of the endothelium are closely related to the comorbidities of atherosclerosis, nephropathy, and retinopathy that occur in patients with type 1 DM and T2DM [148]. An association of hyperglycemia levels with microvascular complications has been demonstrated in studies of type 1DM and T2DM. Endothelial cell apoptosis is one of the mechanisms related to endothelial damage caused by hyperglycemia [149]. However, the complete understanding of the mechanism of damage caused by hyperglycemia to the endothelium is not fully understood. Evidence shows that insulin resistance and endothelial dysfunction progress in obesity and T2DM. However, other physiological factors contributing to the progression of tissue damage and insulin resistance must be considered together [150]. In addition to hyperlipidemia, other independent factors that threaten endothelial integrity can be found, such as age, low-grade inflammation, hyperlipidemia, and hypertension. These physiological conditions are characteristic of metabolic syndrome and often shape the development of T2DM. The challenge to understanding the connection of all these processes in developing endothelial dysfunction lies in distinguishing whether they are causes or consequences of DM syndrome [151].

The TLR-mediated signaling through nuclear factor kappa B (NF-kB) may be involved in the development of DM [152]. Complications of T2DM and some pathological mechanisms of immune diseases are associated with altered TLR2 expression [153]. Hyperglycemia of retinal ganglion cells increases the expression of TLR2 and TLR4, leading to the expression of proinflammatory molecules localized in the retinal area [154]. The connection of autophagy and inflammation with TLR signaling in β-cell dysfunction through activating proinflammatory transcription factors via MyD88-NF-kB has also been described [155]. Increased FFA promotes TLR4 expression and activation, triggering insulin resistance [156]. Recent studies reported that TLR2 may promote the pathogenesis of type 2 DM. In T2DM and type 1 DM, an increase in the expression and activity of TLR2 and TLR-4 is found. This increase in TLR2 and TLR4 activity is accompanied by a pro-inflammatory state and hyperglycemia [157]. TLR4-mediated insulin resistance can be attributed to the NF-κB and mitogen-activated protein kinases (MAPK) signaling pathway. In T2DM and obesity, the increase in TLR4 activation is due to saturated non-esterified fatty acids and LPS acting as ligands [158]. The activation of TLR2 and TLR4 causes an increase in pro-inflammatory molecules in T2DM patients [108]. TLR-3 is also a molecule that is active in the mechanisms of obesity-induced inflammation. However, TLR3 expression is not associated with the degree of insulin sensitivity or BMI [159].

### 4.13. TLR in NAFLD in Obesity

NAFLD is a pathological fat accumulation disorder of the liver. Metabolic syndrome and obesity are associated with the development of NAFLD. It is estimated that about 25% of people with DM have NAFLD. In addition, NAFLD occurs in >50% of people with DM [160,161]. NAFLD occurs more frequently in men. The worldwide incidence may be as high as 47 cases per 1000 [162]. NAFLD presents a range of severity of liver damage that may begin with minimal inflammation from hepatic fatty infiltration and then develop into nonalcoholic steatohepatitis (NASH), liver fibrosis, and ultimately cirrhosis [163]. NASH is a major risk factor in patients with NAFLD for the development of cirrhosis and hepatocellular carcinoma [164]. In addition to liver complications, patients with NAFLD have insulin resistance, dyslipidemia, and risk of [165].

TLRs are transmembrane type I glycoprotein receptors that recognize PAMPs via their leucine-rich repeat ectodomain [166]. The binding of PAMPs to TLRs activates the Myd88 and FN β-inducible adaptor signaling cascades through the TIR (TRIF)/TRIF-related adaptor molecule (TRAM) domain [167]. TLRs respond to endogenous and exogenous ligands derived from parasitic, viral, fungal, and bacterial antigens [168]. All ten different types of TLRs are expressed in the human liver [169]. Bacterial toxins and intestinal barrier permeability are associated with impaired hepatic TLR expression in people and mice with NAFLD [170]. Patients with NALFD have reduced bacterial diversity, termed “intestinal dysbiosis,” in clinical studies [171]. Studies suggest that altered gut microbiota might lead to hepatic fat accumulation through several mechanisms, including increased translocation of bacterial toxins that promote low-grade inflammation and changes in intestinal permeability [172]. These observations contribute to the concept of metabolic inflammation as an underlying modification leading to the hepatic phenotype that is linked to extrahepatic morbidity in NALFD [173].

The ligands for hepatic TLRs can be lipopolysaccharide (LPS) from intestinal bacteria, fatty acids, and high mobility group protein B1 [HMGB1] from damaged hepatocytes. The development of NASH, ASH, and fibrosis or liver cancer is linked to the activation of TLR4 by LPS [174]. Various TLRs could induce gene activations through the same pathway or a different signal transduction pathway within minutes of pathogen invasion, resulting in cellular expression of pro-inflammatory cytokines and chemokines, positive regulation of co-stimulatory factor, and increased antigen presentation capacity, causing a systemic inflammatory response and the emergence of the innate immune response [175]. TLR4 is an LPS receptor and elevation of LPS in most animal models of NAFLD causes hepatic steatosis, hepatic insulin resistance, and increased liver weight, making TLR4-LPS the key pathway promoting the development of NAFLD [176]. TLR9 is associated with the development of NASH. Without TLR9, rats have less steatohepatitis, insulin resistance, and fibrosis [177]. The adaptor protein MyD88 is involved in signaling the expression of proinflammatory molecules mediated by TLRs except TLR3. This TLR-MyD88 signaling promotes the activation of the JNK and NF-κB signaling pathways, which are also related to NAFLD comorbidity [178]. In particular, TLR4 requires the collaboration of two co-receptors with leucine-rich domains, CD14 and MD-2. TLR4 activation by LPS depends on the co-receptor MD-2 [179]. MD-2 also plays a role in the expression of TLR4 on the cell surface and in activating signaling cascades [180]. The function of CD14 is to facilitate the binding of the LPS antigen to the MD-2-TLR-4 complex. This complex induces a conformational change that allows homodimerization with a second TLR4 [181]. The binding of TLR2 with ligands also involves the presence of a co-receptor, CD36. However, the participation of CD36 in TLR2 signaling is still unclear [182].

The liver is considered a first-line defense organ because it filters different intestinal molecules, including exogenous products of bacteria [183]. Most of the macrophages of the gut are located in the liver and are called Kupffer cells. These Kupffer cells are responsible for the identification and phagocytosis of endogenous and exogenous toxins, including bacteria, through the portal vein [184]. Under normal conditions, the liver has a good tolerance to exogenous molecules derived from gut bacteria because it presents low levels of each type of TLR mRNA [185]. The accumulation of hepatic fat has been related to different properties of the gut microbiota. The properties of the cecal microbiota are different in obese humans and mice compared to lean controls [186]. Obesity shows an alteration in the capacity of the cells of the intestinal microbiota to harvest energy and changes in the distribution of all phylogenetic types of gut bacteria [187].

Changes in the gut microbiota may affect intestinal permeability and increase the absorption of microbial LPS associated with the development of NASH [188]. Experiments show that administration of high doses of LPS increases sensitivity to endotoxin hepatotoxicity and subsequent development of steatohepatitis in obese mice [189]. Liver injury from LPS endothelial damage can be ameliorated by impairment of TLR4 expression in the NASH model [190]. On the other hand, TLR2 deficiency does not confer protective effects for steatohepatitis [191]. The mechanisms of NAFLD are mainly related to TLR4-LPS signaling and not necessarily to the peptidoglycan ligands of TLR2 [113]. Regulation of adequate gut microbiota is a promising target for preventing and treating NAFLD [192].

Other effects related to TLR signaling in NAFLD are the activation of ROS-forming mechanisms that favor the release of plasminogen activator inhibitor-1 (PAI-1) in the liver and the development of insulin resistance [169]. In patients with NAFLD, only an increase in the expression of TLRs 1-5 was found compared to the control group. Meanwhile, TLRs 6-10 expression is similar in patients with and without NAFLD [169]. Furthermore, the TLR1-5 expression is accompanied by an increase in MyD88 in these same NAFLD patients. The pathogenic mechanisms of NAFLD are linked to the capacity of TLR1-5 to activate the signaling pathways of PAI-1 and MyD88 [169].

Multivariate risk factors contribute to the development of NALFD, including aging, an unbalanced and high-calorie diet, and low physical activity [193]. Lifestyle management is the first-line treatment to attenuate or reverse NALFD. Dietary changes, weight loss, and moderate to intense aerobic and resistance-type physical exercise contribute to reducing the risk of developing NALFD [194].

Omega-3 fatty acids (ω3) can be considered a non-pharmacological option to help control the inflammatory effect of NALFD [195]. ω3 belongs to a family of polyunsaturated fatty acids (PUFA), and its intake in supplements is associated with a decreased risk of coronary events and heart disease [196]. PUFA has been shown to play a role in regulating proinflammatory processes. The anti-inflammatory mechanisms of ω3 are related to the modulation of the immune response of B and T lymphocytes and the decreased release of proinflammatory cytokines such as TNF-alpha and IL-2 [197]. ω3 supplementation has been shown to benefit weight control and cognitive performance in obese subjects [198]. In addition, ω3 has also helped improve liver markers in patients with NAFLD [199]. Anti-inflammatory mechanisms of ω3 PUFA may also be linked to TLR signaling. In a macrophage culture model, it was shown that the administration of PUFAs, including ω3, activates TLR4 and causes the release of 22-carbon fatty acids that inhibit cyclooxygenase-mediated inflammatory processes [200].

Pharmacologic therapy for NAFLD requires medical evaluation and needs to be accompanied by nonpharmacologic interventions. Drug-based therapies to treat NALFD are still limited [201]. Weight loss can improve NAFLD, NASH, and fibrosis, and the benefits and effects of weight control medications for NAFLD control should be studied. Vitamin E has been shown to be an anti-inflammatory drug that protects liver tissue in NALFD compared with a placebo, according to a randomized study of people with NALFD without diabetes [202].

### 4.14. TLR in Cancer and Obesity

Cancer is a syndrome related to a group of diseases in which the central pathological process lies in the abnormal and uncontrolled growth of cells in an organ or tissue. The human body carefully regulates the cell growth of each organ and tissue by administering nutrients and macromolecules necessary for cell proliferation [203]. The body’s quantity and availability of nutrients are a factor in controlling the systemic signaling networks of cell growth. The transformation of neoplastic cells can be favored by activating growth factors mediated by the chronic excess of nutrients stored in obese subjects [203]. The increase in energy over-storage makes obesity a risk factor for tumors of different organs and tissues such as the colon, pancreas, prostate, and breast cancer. Patients suffering from cancer and obesity have a higher risk of mortality compared to lean cancer patients [204]. Although BMI is an excellent tool used in the clinic to detect excess body mass, it cannot distinguish the distribution of mass between each type of tissue, which is an important factor in identifying neoplastic tissue [205]. Today, there are image processing software that measure AT and skeletal muscle content from CT images. These software are used to track tumor progression in cancer patients to determine the efficacy of treatments or the speed with which cancer develops [206]. The metabolic processes of AT are mediated by signaling pathways responsible for the body’s energy balance. The communication of available nutrients between cells, tissues, and organs is carried out through a hormonal signaling network. An example of this communication system is glucose homeostasis, managed by the interaction between AT, skeletal muscle, kidneys, liver, and pancreas [206]. Insulin is the most important hormone for regulating systemic energy in the body. Insulin regulates the concentration of circulating glucose, cellular glucose uptake, and the production of proteins and energy to execute the cell cycle [207]. This endocrine system regulates cell proliferation and growth in normal conditions but not in cancer. In cancer cells, gene mutations express correct cell proliferation mechanisms and impair cellular metabolism, causing a hyperproliferative state of high energy demand [208]. Aerobic glycolysis tends to predominate in cancer cell metabolism as a source of ATP rather than oxidative phosphorylation. It is proposed that aerobic glycolysis metabolism provides a rapid activation of energy production and that the enzymes necessary for this also contribute to the biosynthesis of cell proliferation [208]. The increased lipid storage in obesity changes cellular metabolism and establishes a state that communicates to the cells a state of excess nutrients. Chronic over nutrition triggers signaling cascades so that the cells increase glucose intake and execute the mechanisms of cell growth and proliferation, which increases the risk of neoplastic transformation [209]. Some specific mutations in cancer cells can condition their energy metabolism in obesity. Cancer cells with mutations in phosphatidylinositol 3-kinase (PI3K) signaling to adapt to caloric restriction [210]. Cancer cells may obtain the energy resources and substrates necessary for dividing their cellular organelles in abundance in obesity. Furthermore, some cancer cells can adapt to beta-oxidation-driven energy metabolism [211]. Beta-oxidation-dependent cells are especially favored in an environment of FFA over storage [212]. The vast lipid storage in the visceral region is a source of large amounts of energy that can sustain uncontrolled cell growth of cancer cells [213]. Targeting the fatty acid entry pathway into the mitochondria may be a helpful treatment against beta-oxidation-dependent cancer cells in cellular models of lung and prostate cancer [214]. It is also observed that adiponectin may have a protective effect against cancer due to its ability to inhibit cell growth in some experimental models of colon and endometrial cancer [215]. Adiponectin has the property of inducing cellular arrest through adenosine monophosphate-activated protein kinase signaling and mTOR inhibition. Adiponectin is often decreased in obese subjects, which may add to the risk of developing tumor growth. Leptin is a molecule that has anti-adiponectin effects so measurement of the adiponectin/leptin ratio may serve as a marker linking obesity and cancer [216]. Other molecules with effects that promote cell growth are estrogens and androgens. In men and postmenopausal women, the production of estrogens from androgens occurs in AT [217]. The activity of estrogens on cells is predominantly proliferative and inhibits apoptosis. Obesity has been observed to be a risk factor for breast cancer in post-menopausal women [218].

In addition, obesity decreases the concentration of the protein globulin transporter, which has been associated with an increase in free estradiol and the risk of cancer in postmenopausal women [219]. Due to the wide variety of oncogenic factors that are triggered by obesity, weight control management should be an essential task in the care of patients with cancer. Evidence shows that weight loss in patients with obesity reduces the propensity for cancer. Exercise and a low-calorie diet are the most effective strategies for weight loss [220]. A controlled low-carbohydrate (ketogenic) diet may be effective in improving insulin sensitivity, weight loss, and lowering glucose [221]. The benefits of diets for weight loss and lowering blood glucose and cholesterol may contribute to reducing tumor proliferation signaling. However, more evidence is still required to detail the characteristics that diets should have to obtain clear clinical benefits for cancer [222]. The relationship between obesity and immune-inflammatory processes in cancer has led to attention to TLRs. Therapeutic proposals related to TLRs in cancer mainly focus on suppressing TLRs inflammatory and energy-damaging effects. On the other hand, they also focus on activating and modulating the immune-mediated effects of TLRs to combat pathogens [216]. The challenge for a therapy to control cancer cells lies in determining the degree of accurate TLR activation necessary for cancer cell elimination [223].

Although the hypothesis of using TLRs to treat cancer has a feasible beneficial effect, it is also true that such therapy can bring important adverse effects. An inadequate regulation in the signaling of TLRs can lead to a detrimental activation of the immune system and trigger autoimmune diseases. Also, the over-activation of TLRs can worsen the comorbidities associated with the chronic inflammatory state [224]. On the other hand, the suppression of TLRs can promote a state of immune tolerance and increase the risk of establishing opportunistic diseases [225]. Accurate therapeutic protocols for administering TLR modulators are essential, and a great deal of work is required for the standardization of the doses of TLR agonist and antagonist agents for the correct management of the safety of patients with cancer. Information in this regard is in the early stages Table 1.

## 5. Potential Therapies

The healthcare system should consider the treatment of obesity with empathy and without prejudice. Lifestyle changes such as low-calorie diets and physical activity should be considered as the first line of treatment [221]. The use of obesity control medications should be used to control the risk of obesity-related comorbidities. For weight loss treatments to have a long-term effect, they must generate a marked cognitive-behavioral lifestyle in patients [194]. Bariatric surgery is also a therapeutic option for treating patients with severe obesity. However, treatment of obesity should be individualized, considering the demographic characteristics of patients such as gender, age, BMI, individual health risks, comorbidities, and psycho-behavioral characteristics. Individualization of weight loss therapy should also consider genetic factors [226]. The following figure briefly groups the impact of obesity on related diseases Figure 2.

## 6. Conclusions

Obesity is a significant public health problem affecting developed and even developing societies. Obesity increases the risk of highly prevalent diseases such as CVD, insulin resistance, T2DM, NAFLD, and cancer. Modulating TLR activity through diet, exercise, and pharmacological agents is being explored as a possible therapeutic approach to controlling obesity and its complications. TLR activation produces proinflammatory cytokines (TNF-α, IL-6, and IL-1β) that are essential in developing insulin resistance. TLRs are involved in the pathogenesis of type 2 DM. Chronic inflammation driven by TLR activation alters insulin signaling, leading to insulin resistance. In addition, TLR2 and TLR4 activation can lead to endothelial dysfunction and vascular complications associated with diabetes. TLR4 is a key factor in NAFLD. It is activated by fatty acids and LPS from the gut microbiota, promoting the production of inflammatory cytokines. TLRs are actively involved in the immune system’s response to cancer. They can recognize cancer cell-derived molecules and activate immune responses. The role of TLRs in cancer is dual: (a) Cancer immunity: activation of certain TLRs may enhance the immune system’s ability to recognize and destroy cancer cells, and (b) Pro-cancer effects: chronic inflammation driven by TLR activation may create an environment that favors tumor growth and metastasis. The global organ involvement in obesity is an unfinished business, where the epidemiological role is fundamental to avoiding obesity-related metabolic disorders.

## Figures and Tables

**Figure 1 ijms-26-02229-f001:**
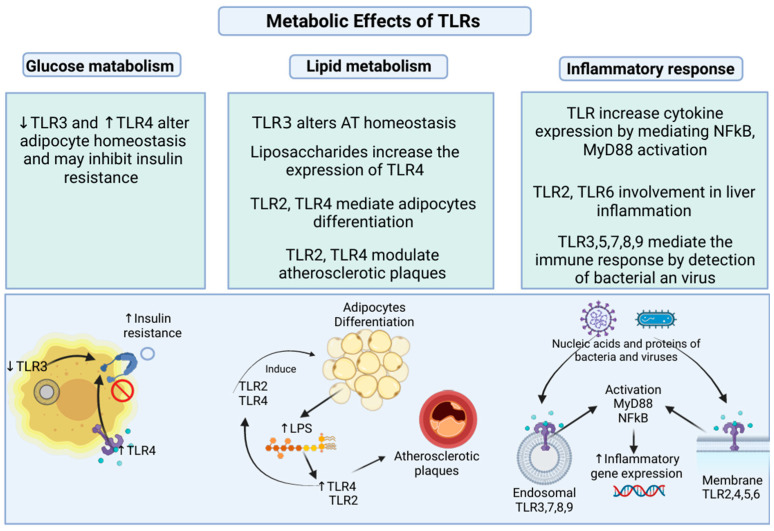
Metabolic effects of TLRs. Decreased TLR3 and increased TLR4 are involved in cellular glucose metabolism by increasing the insulin resistance. Adipose tissue (AT) homeostasis is associated with TLR3 expression [55]. TLR4 may promote insulin resistance [56]. The increase of TLR2 and TLR4 modulate in adipocyte differentiation [57]. TLR4 modulates lipid metabolism in AT and is upregulated in atherosclerotic plaques together with TLR2 [58]. Membrane and endosomal TLRs upregulate cytokines and other inflammatory genes in response to bacterial and viral antigen identification [42]. Created in BioRender. Garcia, A. (2025). https://BioRender.com/m42i307 (accessed on 28 February 2025).

**Figure 2 ijms-26-02229-f002:**
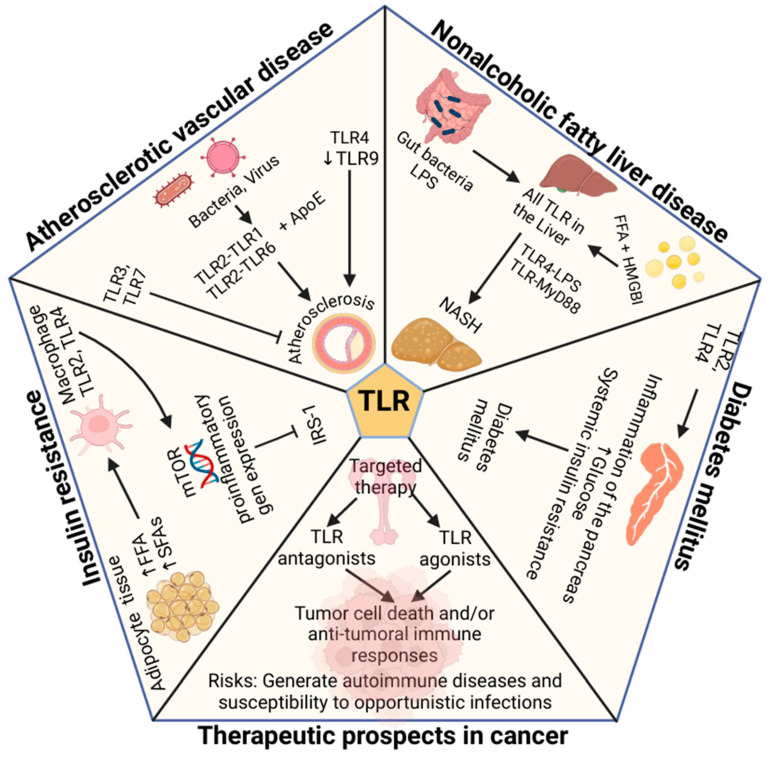
Scheme of possible interactions of TLRs in different metabolic disorders related to obesity. Bacteria or viruses in atherosclerotic plaques can activate TLRs [120]. TLR2-TLR1 and TLR2-TLR6 heterodimers are associated with atherosclerotic plaques [125]. TLR4 and decreased TLR9 are also related to the development of atherosclerotic plaques, while TLR3 and TLR7 could act as protective factors [86,128,129,130]. Increased free fatty acids (FFA) and saturated fatty acids (SFAs) trigger macrophage activity via TLR2 and TLR4 [41,137]. Macrophage activity triggers mTOR signaling, favoring the expression of pro-inflammatory molecules, leading to inactivation of insulin receptor substrate 1 (IRS-1) and insulin resistance. TLR2 and TLR4 receptors are elevated in immune diseases such as type 1 diabetes and are associated with beta-cell dysfunction and insulin resistance [60,155]. Studies show that bacterial lipopolysaccharides (LPS) derived from intestinal microflora, endogenous molecules such as high mobility group protein B1 [HMGB1] and FFA promote fat accumulation and liver inflammation via TLR4-LPS and TLR9 [178,179]. Anti-cancer TLR research focuses on activating or suppressing their immunomodulatory effects to enhance the immune response against cancer cells [224]. Created in BioRender. Garcia, A. (2025) https://BioRender.com/e62d130 (accessed on 28 February 2025).

**Table 1 ijms-26-02229-t001:** Representative research studies on TLR and metabolic disease models.

Studied TLR	Disease Model	Method	Results	Cite
TLR1, TLR2, TLR6	Atherosclerosis	TLR1- or TLR6-deficient mice fed a high-fat diet (HFD) for 10 weeks.	TlR2 interacts with either TLR1 or TLR6 to promote atherosclerosis in LDL receptors. Deficiency of TLR1 or TLR6 did not diminish HFD-driven disease.	[125]
TLR4	Atherosclerosis	TLR4 and LDL receptor double knockout (*TLR4*^−/−^*Ldlr*^−/−^) mice and *Ldlr*^−/−^ mice were fed either a standard chow or a diabetogenic diet for 24 weeks.	TLR4 deficiency markedly decreased atherosclerosis in obese *Tlr4*^−/−^*Ldlr*^−/−^ mice.	[126]
TLR3	Atherosclerosis	Hypercholesterolemia-induced arterial injury in mice and human atheroma-derived smooth muscle cells.	Genetic deletion of TLR3 dramatically enhanced the development of elastic lamina damage after collar-induced injury. Deficiency of TLR3 accelerated the onset of atherosclerosis.	[128]
TLR9	Atherosclerosis	Double-knockout ApoE^−/−^:TLR9^−/−^ mice and control ApoE^−/−^ mice were fed a high-fat diet from 8 weeks.	Genetic deletion of the innate immune receptor TLR9 exacerbated atherosclerosis lesion severity.	[130]
TLR2, TLR4	Diabetic retinopathy	Human retinal ganglion cell culture (control, high-glucose, and siRNA-transfected groups).	TLR2/4, TNF-α, and IL-8 were significantly increased in retinal ganglion cells treated with high-glucose.	[156]
TLR4	Diabetes mellitus type 1	Healthy controls (*n* = 37) and type 1 diabetic patients (*n* = 34) were recruited, and a fasting blood sample was obtained.	TLR4 ligand and endotoxin were significantly elevated in type 1 diabetic patients compared to matched controls. Hsp60 and HMGB1 concentrations were also considerably increased in the patients.	[159]
TLR4	Non-alcoholic steatohepatitis	Male control and TLR4 mutant mice with fed control or methionine/choline-deficient diet (MCDD).	Histological evidence is typical of steatohepatitis, portal endotoxemia, and enhanced TLR4 expression in wild-type mice fed MCDD.	[192]
TLR 1-5	Non-alcoholic steatohepatitis	11 patients with NAFLD that underwent either a partial liver resection or 11 controls.	Livers of NAFLD patients had significantly higher Hepatic TLR 1-5 mRNAs expression, increased lipid peroxidation, and alterations in insulin signaling.	[169]
TLR9	Non-alcoholic steatohepatitis	Type Mice and mutant TLR9 mice were steatohepatitis induced by a choline-deficient amino acid-defined diet.	Mutant TLR9 mice showed less steatohepatitis and liver fibrosis than wild-type mice.	[179]
TLR4	Non-alcoholic steatohepatitis	Male control and TLR4, TLR2 mutant mice with steatohepatitis induced.	TLR4 mutant mice had lower injury and lipid accumulation markers than control.	[194]
TLR4, TLR2	Non-alcoholic steatohepatitis	Male control and TLR4, TLR2 mutant mice with nonalcoholic steatohepatitis-induced.	TLR4 wild-type and TLR2 deficient mice had more liver injury in nonalcoholic steatohepatitis.	[193]

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
