# Peer review of "The Role of TLRs in Obesity and Its Related Metabolic Disorders"

_ijms, 2025, doi:10.3390/ijms26052229_

Round 1
Reviewer 1 Report (Previous Reviewer 1)
Comments and Suggestions for Authors
Many scientific descriptions need proper citations, please revise them.
Author Response
REVIEWER 1
Comment: The scientific writing was incompleted in the current form of this manuscript. This manuscript's style in reference citation was incorrect and did not fit the guidelines of IJMS. Many important scientific descriptions only cite one reference in this manuscript.
Answer: I appreciate your comments and will respond to them. The document was reviewed and the citations were placed in the corresponding paragraphs and sentences
Comment: Additionally, please check the similarity of this manuscript by using Turnitin. I vaguely remember that the similarity of this manuscript is over 60% (that is not acceptable for article publication). If so, this manuscript should be rejected.
Answer: Detailed work was carried out on the manuscript to improve the English writing and to detect and change the sentences that contribute to the percentage of similarity

Reviewer 2 Report (New Reviewer)
Comments and Suggestions for Authors
The review results for this study are as follows.
Since this study is a review, the process of selecting prior studies for review is crucial. However, this study does not mention the research methodology at all. The research methodology must be explicitly stated, The research methodology must be explicitly stated, including IRB approval. It should specify whether a systematic review or a meta-analysis was conducted, and whether specific methods such as PRISMA were applied. The inclusion and exclusion criteria for the reviewed studies must also be clearly defined. Additionally, the selection process of the studies included in this review should be presented in a flow diagram. The final number of analyzed studies, the theoretical basis for deriving key indicators such as TLRs, and the conclusions should be drawn based on the analysis results of the reviewed studies.
Author Response
REVIEWER 1
Comment: The scientific writing was incompleted in the current form of this manuscript. This manuscript's style in reference citation was incorrect and did not fit the guidelines of IJMS. Many important scientific descriptions only cite one reference in this manuscript.
Answer: The document was reviewed and the citations were placed in the corresponding paragraphs and sentences
Comment: Additionally, please check the similarity of this manuscript by using Turnitin. I vaguely remember that the similarity of this manuscript is over 60% (that is not acceptable for article publication). If so, this manuscript should be rejected.
Answer: Detailed work was carried out on the manuscript to improve the English writing and to detect and change the sentences that contribute to the percentage of similarity
I appreciate your comments, I hope I have corrected what was requested.

Reviewer 3 Report (New Reviewer)
Comments and Suggestions for Authors
Thank you for your submitting precise review paper to this journal.
This review paper is very clear and informative to all researchers and related persons to understand the association between TLRs(inflammation mediators) and many diseases(metabolic diseases and cancer), especially, to understand the mechanisms of various diseases developments.
I will give minor word errors in line of 42, "most parts of the ??"
in line of 156, "umans"
in line of 196, "LR4"
in line of 375, need to change into English
in line of 607, need to change into English
in line from 636 to 642, need to change into English
Comments on the Quality of English Language
Same as above
Author Response
REVIEWER 2
Comment: This file records traces of changes. Please organize the files in an orderly fashion.
Answer: We appreciate the comments on improving the writing on behalf of the authors who contributed to the document.
Best regards
Alejandra Guillermina Miranda-Díaz, MD, PhD

Reviewer 4 Report (New Reviewer)
Comments and Suggestions for Authors
Manuscript ID: ijms-3470798
The Role of TLRs in Obesity and its Related Metabolic Disorders
It is important to address how lifestyle changes through diet and exercise down-regulation the gene and protein expression of TLRs in certain pathologies, as well as the use of certain drugs and bariatric surgery in established obesity and in different degrees of obesity. You mentioned that these would be the objectives of the manuscript.
Line 42 in the paragraphs the idea is unfinished, because did you write “they encourage a more stressful and sedentary lifestyle with unhealthy eating habits in most parts of the [1]”
Line 45 you need to mention which organic systems are affected by obesity.
Line 60 Could you mention what the others are energy-producing metabolic processes?
Line 60-64 what are the principal functions of Adipokines? how this proteins participate in the insulin signaling regulated, glucose uptake, fatty acid oxidation and other energy-producing metabolic processes. you need write what is the rol of the adipokines and citokines, example: proinflammatory or anti-inflammatory, what are the inflammatory or anti-inflammatory citokines or adipokines?
Line 65-67 It is necessary to describe how excessive caloric intake is associated with increased adipocity and inflammation. What type of inflammation is being discussed, acute or chronic?. could you describe how it develops the inflammation you are mentioning . You could use a diagram to represent it.
Line 71 – 141 Inflammation in obesity (2 point)
You need to describe in detail how the low-grade inflammatory process that occurs in obesity is generated. It is necessary to define which cells participate in the generation of the low-grade inflammatory process. What is the role of adipocytes in this low-grade inflammatory process? Are only macrophages involved in the secretion of cytokines? What is the result obtained by the low-grade inflammatory process on adipocytes, hypertrophy or hyperplasia? Could you mention which cytokines and adipokines are secreted by macrophages and which by adipocytes. Mention what type of communication is generated between these two classes of cells in adipose tissue (macrophages and adipocytes). Mention and describe which macrophage phenotype predominate in adipose tissue.
Line 178-180 What are the cells that participate in the generation of these cellular responses such as the production of cytokines and chemokines responsible for the inflammatory response and the immune attack against infection.
Ljne 222 what type of cell membranes does heterodimerization between TLR1 and TLR2 occur? ¿In which cells does it mediate immune and innate responses?
Line 240-241 you said "Inflammatory stimulation of TLR2 induces expression changes in secretory cytokines such as IL-6" in this case, what is the role of IL-6 cytokine, inflammatory or anti-inflammatory citokyne?
Line 251-253 you said “Deletion of TLR2 results from preservation of postischemic coronary endothelial function and prevention of abnormal ventricular remodeling” Could you explain this process in more detail?
312-317
Could you show a diagram of the TLR3 and TLR4 signaling pathway in a cell and in what state these proteins are present in cell membranes and when they are overexpressed due to the presence of a pathology or an active pathological state?
334-339 Could you mention the signaling cascade generated by the activation of TLR5 and add more information on this please?
Line 345 you said "TLR en enfermedad vascular aterosclerótica" you need to change this line in English
Line 421-422 in this paragraph "TLR2-TLR1 heterodimer and TLR2-TLR6 heterodimer contribute to atherosclerosis 421
in ApoE gene-deficient mice (ApoE-/-) and LDL receptor-deficient mice (LDLR-/-) [ ]" the reference has been lost.
Line 448-463 could you mention what happens with the density of insulin receptors present in the different cell lines, when there is insulin resistance?
Line 636- 642 this section is in Spanish “Estudios sugieren que el microbiota intestinal alterada podría conducir a la acumulación de grasa hepática a través de varios mecanismos, incluida mayor translocación de toxinas bacterianas que promueven inflamación de bajo grado y cambios en la permeabilidad intestinal [172]. Estas observaciones contribuyen al concepto de inflamación metabólica como una modificación subyacente que conduce al fenotipo hepático que está vinculada a la morbilidad extrahepática en NALFD”.you need to change .
In all subtitles, the relationship of TLRs with the different pathologies mentioned should be integrated more forcefully.
Line 607 – 756 the point 3.12 and 3.13 they are lost in the manuscript.
Comments on the Quality of English LanguageThe English could be improved to more clearly express the research.
Author Response
REVIEWER 2
This paper discussed the role of (TLRs) in obesity. Therefore, we believe it will help understand the role of TLRs in obesity and obesity-related metabolic diseases. It provides a variety of information by analyzing a wide range of previous studies and provides valuable information overall.
Comment: However, discussion in the recent literature seems to be insufficient. I request that you analyze and add recent research results.
Answer: A detailed review of the entire manuscript was conducted, and improvements were made to the English language and the explanation of the topics. The primary information focused specifically on the main topic of the manuscript is detailed, but changes were noted in the citations to improve the explanation of the topics. Some of the specific changes are highlighted in yellow
Comment: Please add pictures (figures) to explain some important points more easily. Request to check the numbering of references.
Answer. Two images were created that summarize the main topics of the manuscript related to the characteristics of TLRs and their effects on different metabolic diseases associated with obesity. Images can be modified if there is any suggestion to improve them
Figure 1. METABOLIC EFFECTS OF TLRS. TLR3 and TLR4 are involved in cellular glucose metabolism. Adipose tissue (AT) homeostasis is associated with TLR3 expression [78]. TLR4 may promote insulin resistance [80]. TLR2 and TLR4 modulate in adipocyte differentiation [70]. TLR4 modulates lipid metabolism in AT and is upregulated in atherosclerotic plaques together with TLR2 [63]. Membrane and endosomal TLRs upregulate cytokines and other inflammatory genes in response to bacterial and viral antigen identification [42]. Created in BioRender. Garcia, A. (2025) https://BioRender.com/m42i307
Figure 2. SCHEME OF POSSIBLE INTERACTIONS OF TLRS IN DIFFERENT METABOLIC DISORDERS RELATED TO OBESITY. Bacteria or viruses in atherosclerotic plaques can activate TLRs [120]. TLR2-TLR1 and TLR2-TLR6 heterodimers are associated with atherosclerotic plaques [125]. TLR4 and decreased TLR9 are also related to the development of atherosclerotic plaques, while TLR3 and TLR7 could act as protective factors [86,128-130]. Increased free fatty acids (FFA) and saturated fatty acids (SFAs) trigger macrophage activity via TLR2 and TLR4 [41,137]. Macrophage activity triggers mTOR signaling, favoring the expression of pro-inflammatory molecules, leading to inactivation of insulin receptor substrate 1 (IRS-1) and insulin resistance. TLR2 and TLR4 receptors are elevated in immune diseases such as type 1 diabetes and are associated with beta-cell dysfunction and insulin resistance [60,155]. Studies show that bacterial lipopolysaccharides (LPS) derived from intestinal microflora, endogenous molecules such as high mobility group protein B1 [HMGB1] and FFA promote fat accumulation and liver inflammation via TLR4-LPS and TLR9 [178,179]. Anti-cancer TLR research focuses on activating or suppressing their immunomodulatory effects to enhance the immune response against cancer cells [224]. Created in BioRender. Garcia, A. (2025) https://BioRender.com/e62d130.

Round 2
Reviewer 1 Report (Previous Reviewer 1)
Comments and Suggestions for Authors
No further comment.
Author Response
REVIEWER 1
Comment: The scientific writing was incompleted in the current form of this manuscript. This manuscript's style in reference citation was incorrect and did not fit the guidelines of IJMS. Many important scientific descriptions only cite one reference in this manuscript.
Answer: I appreciate your comments and will respond to them. The document was reviewed and the citations were placed in the corresponding paragraphs and sentences
Comment: Additionally, please check the similarity of this manuscript by using Turnitin. I vaguely remember that the similarity of this manuscript is over 60% (that is not acceptable for article publication). If so, this manuscript should be rejected.
Answer: Detailed work was carried out on the manuscript to improve the English writing and to detect and change the sentences that contribute to the percentage of similarity
REVIEWER 1
Comment: Many scientific descriptions need proper citations, please revise them.
Answer: I appreciate the comment to improve the manuscript. The document was reviewed and the citations were placed in the corresponding paragraphs and sentences

Reviewer 2 Report (New Reviewer)
Comments and Suggestions for Authors
This study is a review research, and the research methodology is very important. However, there is no mention of the research methodology at all. The research topic, "The Role of TLRs in Obesity and its Related Metabolic Disorders," should be supported by an analysis of RCTs (Randomized Controlled Trials) with a rigorous research methodology. Additionally, the selection of performance indicators related to the research topic and the rationale behind them have not been provided. Furthermore, the references are not formatted according to the journal's guidelines.
Author Response
REVIEWER 2
This paper discussed the role of (TLRs) in obesity. Therefore, we believe it will help understand the role of TLRs in obesity and obesity-related metabolic diseases. It provides a variety of information by analyzing a wide range of previous studies and provides valuable information overall.
Comment: However, discussion in the recent literature seems to be insufficient. I request that you analyze and add recent research results.
Answer: A detailed review of the entire manuscript was conducted, and improvements were made to the English language and the explanation of the topics. The primary information focused specifically on the main topic of the manuscript is detailed, but changes were noted in the citations to improve the explanation of the topics. Some of the specific changes are highlighted in yellow
Comment: Please add pictures (figures) to explain some important points more easily. Request to check the numbering of references.
Answer. Two images were created that summarize the main topics of the manuscript related to the characteristics of TLRs and their effects on different metabolic diseases associated with obesity. Images can be modified if there is any suggestion to improve them
Figure 1. METABOLIC EFFECTS OF TLRS. TLR3 and TLR4 are involved in cellular glucose metabolism. Adipose tissue (AT) homeostasis is associated with TLR3 expression [78]. TLR4 may promote insulin resistance [80]. TLR2 and TLR4 modulate in adipocyte differentiation [70]. TLR4 modulates lipid metabolism in AT and is upregulated in atherosclerotic plaques together with TLR2 [63]. Membrane and endosomal TLRs upregulate cytokines and other inflammatory genes in response to bacterial and viral antigen identification [42]. Created in BioRender. Garcia, A. (2025) https://BioRender.com/m42i307
Figure 2. SCHEME OF POSSIBLE INTERACTIONS OF TLRS IN DIFFERENT METABOLIC DISORDERS RELATED TO OBESITY. Bacteria or viruses in atherosclerotic plaques can activate TLRs [120]. TLR2-TLR1 and TLR2-TLR6 heterodimers are associated with atherosclerotic plaques [125]. TLR4 and decreased TLR9 are also related to the development of atherosclerotic plaques, while TLR3 and TLR7 could act as protective factors [86,128-130]. Increased free fatty acids (FFA) and saturated fatty acids (SFAs) trigger macrophage activity via TLR2 and TLR4 [41,137]. Macrophage activity triggers mTOR signaling, favoring the expression of pro-inflammatory molecules, leading to inactivation of insulin receptor substrate 1 (IRS-1) and insulin resistance. TLR2 and TLR4 receptors are elevated in immune diseases such as type 1 diabetes and are associated with beta-cell dysfunction and insulin resistance [60,155]. Studies show that bacterial lipopolysaccharides (LPS) derived from intestinal microflora, endogenous molecules such as high mobility group protein B1 [HMGB1] and FFA promote fat accumulation and liver inflammation via TLR4-LPS and TLR9 [178,179]. Anti-cancer TLR research focuses on activating or suppressing their immunomodulatory effects to enhance the immune response against cancer cells [224]. Created in BioRender. Garcia, A. (2025) https://BioRender.com/e62d130.
REVIEWER 2
Comment: This study is a review research, and the research methodology is very important. However, there is no mention of the research methodology at all. The research topic, "The Role of TLRs in Obesity and its Related Metabolic Disorders," should be supported by an analysis of RCTs (Randomized Controlled Trials) with a rigorous research methodology.
Answer: Comments made to improve the manuscript are appreciated. Regarding this comment, I agree that systematic reviews are characterized by a thoroughly described methodology that includes much focused research questions, a detailed research strategy, an assessment of the validity of reviewed research and interpretations of the results of research. However, this manuscript was not constructed as a systematic review but rather as a narrative literature review. The objective of this work is to provide context and background information on TLRs in Obesity and its Related Metabolic Disorders. This review seeks to synthesize the information to provide an understanding of the different theoretical perspectives and to identify topics for further research.
Considering the above, a section of the following search strategy used in this review has been included in the manuscript:
- Search strategy
The search for information was carried out using the electronic databases PubMed, Scopus, Google Scholar. The following keywords were used in the search: TLR, TLR1-10, obesity, adipose tissue, metabolic syndrome, metabolic disease, atherosclerosis, endothelial dysfunction, diabetes, insulin resistance, NAFLD, NASH, fibrosis, inflammation, oxidative stress, treatment, therapeutic, clinical trial, disease, mechanism, vascular disease, cancer. The AND/OR connection operators were used between the keywords. The keywords were searched with the “title” and “abstract” filters. The search for information was complemented by the review of the reference lists of the articles found. Aggregate information on general concepts was also searched in parallel. Articles on clinical trials and in vivo and in vitro experimental studies were included. Articles in languages ​​other than English and Spanish were excluded. No performance indicators related to the research topic were carried out.
Comment: Furthermore, the references are not formatted according to the journal's guidelines
Answer: The format of the references was modified to comply with the ACS style guide outlined in the journal's guidelines

Reviewer 4 Report (New Reviewer)
Comments and Suggestions for Authors
Line 1140. 3.13 section it was lost in the manuscript.
Author Response
REVIEWER 1
Comment: Many scientific descriptions need proper citations, please revise them.
Answer: I appreciate the comment to improve the manuscript. The document was reviewed and the citations were placed in the corresponding paragraphs and sentences
REVIEWER 2
Comment: This study is a review research, and the research methodology is very important. However, there is no mention of the research methodology at all. The research topic, "The Role of TLRs in Obesity and its Related Metabolic Disorders," should be supported by an analysis of RCTs (Randomized Controlled Trials) with a rigorous research methodology.
Answer: Comments made to improve the manuscript are appreciated. Regarding this comment, I agree that systematic reviews are characterized by a thoroughly described methodology that includes much focused research questions, a detailed research strategy, an assessment of the validity of reviewed research and interpretations of the results of research. However, this manuscript was not constructed as a systematic review but rather as a narrative literature review. The objective of this work is to provide context and background information on TLRs in Obesity and its Related Metabolic Disorders. This review seeks to synthesize the information to provide an understanding of the different theoretical perspectives and to identify topics for further research.
Considering the above, a section of the following search strategy used in this review has been included in the manuscript:
- Search strategy
The search for information was carried out using the electronic databases PubMed, Scopus, Google Scholar. The following keywords were used in the search: TLR, TLR1-10, obesity, adipose tissue, metabolic syndrome, metabolic disease, atherosclerosis, endothelial dysfunction, diabetes, insulin resistance, NAFLD, NASH, fibrosis, inflammation, oxidative stress, treatment, therapeutic, clinical trial, disease, mechanism, vascular disease, cancer. The AND/OR connection operators were used between the keywords. The keywords were searched with the “title” and “abstract” filters. The search for information was complemented by the review of the reference lists of the articles found. Aggregate information on general concepts was also searched in parallel. Articles on clinical trials and in vivo and in vitro experimental studies were included. Articles in languages ​​other than English and Spanish were excluded. No performance indicators related to the research topic were carried out.
Comment: Furthermore, the references are not formatted according to the journal's guidelines
Answer: The format of the references was modified to comply with the ACS style guide outlined in the journal's guidelines
REVIEWER 3
This review paper is very clear and informative to all researchers and related persons to understand the association between TLRs (inflammation mediators) and many diseases(metabolic diseases and cancer), especially, to understand the mechanisms of various diseases developments.
Comments made to improve the manuscript are appreciated. A response to each comment is provided below:
Comment: I will give minor word errors in line of 42, "most parts of the ??"
Answer: The sentence was corrected and can be seen on line 42
Comment: in line of 156, "umans"
Answer: The word was corrected and can be confirmed on line 112
Comment: in line of 196, "LR4"
Answer: The word was corrected and can be confirmed on line 167
Comment: in line of 375, need to change into English
Answer: The sentences were rephrased and changed into English and can be confirmed in the line 429
Comment: in line of 607, need to change into English
Answer: The sentences were rephrased and changed into English and can be confirmed in the line 635
Comment: in line from 636 to 642, need to change into English
Answer: The sentences were changed to English and can be confirmed in the lines 654 a 659
REVIEWER 4
Comment: Line 1140. 3.13 section it was lost in the manuscript.
Answer: The comment is appreciated. The numbering of the topics was corrected.

Round 3
Reviewer 2 Report (New Reviewer)
Comments and Suggestions for Authors
The second revised manuscript has been appropriately corrected. Thank you for your hard work.
This manuscript is a resubmission of an earlier submission. The following is a list of the peer review reports and author responses from that submission.
Round 1
Reviewer 1 Report
Comments and Suggestions for Authors
The scientific writing was incompleted in the current form of this manuscript. This manuscript's style in reference citation was incorrect and did not fit the guidelines of IJMS. Many important scientific descriptions only cite one reference in this manuscript. Additionally, please check the similarity of this manuscript by using Turnitin. I vaguely remember that the similarity of this manuscript is over 60% (that is not acceptable for article publication). If so, this manuscript should be rejected.
Comments on the Quality of English LanguageThe manuscript needs an English editor.
Author Response
REVIEWER 1
Comment: The scientific writing was incomplete in the current form of this manuscript. This manuscript's style in reference citation was incorrect and did not fit the guidelines of IJMS. Many important scientific descriptions only cite one reference in this manuscript.
Answer: I appreciate your comments and will respond to them. The document was reviewed and the citations were placed in the corresponding paragraphs and sentences
Comment: Additionally, please check the similarity of this manuscript by using Turnitin. I vaguely remember that the similarity of this manuscript is over 60% (that is not acceptable for article publication). If so, this manuscript should be rejected.
Answer: Detailed work was carried out on the manuscript to improve the English writing and to detect and change the sentences that contribute to the percentage of similarity

Reviewer 2 Report
Comments and Suggestions for Authors
This paper discussed the role of (TLRs) in obeê “sity. Therefore, we believe it will help understand the role of TLRs in obesity and obesity-related metabolic diseases. It provides a variety of information by analyzing a wide range of previous studies and provides valuable information overall.
However, discussion in the recent literature seems to be insufficient. I request that you analyze and add recent research results. Please add pictures (figures) to explain some important points in an easier-to-understand manner. Request to check the numbering of references.
Comments on the Quality of English LanguageModerate check of English language is need for effective understanding.
Author Response
REVIEWER 2
This paper discussed the role of (TLRs) in obesity. Therefore, we believe it will help us understand the role of TLRs in obesity and obesity-related metabolic diseases. It provides a variety of information by analyzing a wide range of previous studies and provides valuable information overall.
Comment: However, the recent literature seems to lack sufficient discussion. I request that you analyze and add recent research results.
Answer: A detailed review of the entire manuscript was conducted, and improvements were made to the English language and the explanation of the topics. The primary information focused specifically on the main topic of the manuscript is detailed, but changes were noted in the citations to improve the explanation of the issues. Some of the specific changes are highlighted in yellow
Comment: Please add pictures (figures) to explain some important points more easily. Request to check the numbering of references.
Answer. Two images were created that summarize the main topics of the manuscript related to the characteristics of TLRs and their effects on different metabolic diseases associated with obesity. Images can be modified if there is any suggestion to improve them
Figure 1. METABOLIC EFFECTS OF TLRS. TLR3 and TLR4 are involved in cellular glucose metabolism. Adipose tissue (AT) homeostasis is associated with TLR3 expression [78]. TLR4 may promote insulin resistance [80]. TLR2 and TLR4 modulate in adipocyte differentiation [70]. TLR4 modulates lipid metabolism in AT and is upregulated in atherosclerotic plaques together with TLR2 [63]. Membrane and endosomal TLRs upregulate cytokines and other inflammatory genes in response to bacterial and viral antigen identification [42]. Created in BioRender. Garcia, A. (2025) https://BioRender.com/m42i307
Figure 2. SCHEME OF POSSIBLE INTERACTIONS OF TLRS IN DIFFERENT METABOLIC DISORDERS RELATED TO OBESITY. Bacteria or viruses in atherosclerotic plaques can activate TLRs [120]. TLR2-TLR1 and TLR2-TLR6 heterodimers are associated with atherosclerotic plaques [125]. TLR4 and decreased TLR9 are also related to the development of atherosclerotic plaques, while TLR3 and TLR7 could act as protective factors [86,128-130]. Increased free fatty acids (FFA) and saturated fatty acids (SFAs) trigger macrophage activity via TLR2 and TLR4 [41,137]. Macrophage activity triggers mTOR signaling, favoring the expression of pro-inflammatory molecules, leading to inactivation of insulin receptor substrate 1 (IRS-1) and insulin resistance. TLR2 and TLR4 receptors are elevated in immune diseases such as type 1 diabetes and are associated with beta-cell dysfunction and insulin resistance [60,155]. Studies show that bacterial lipopolysaccharides (LPS) derived from intestinal microflora, endogenous molecules such as high mobility group protein B1 [HMGB1] and FFA promote fat accumulation and liver inflammation via TLR4-LPS and TLR9 [178,179]. Anti-cancer TLR research focuses on activating or suppressing their immunomodulatory effects to enhance the immune response against cancer cells [224]. Created in BioRender. Garcia, A. (2025) https://BioRender.com/e62d130.
I appreciate your comments, I hope I have corrected what was requested.

Round 2
Reviewer 2 Report
Comments and Suggestions for Authors
Overall, I think the points raised have been well corrected. Thank you for your effort
However, this is a file in which traces of modifications are tracked. Please organize the files neatly.
Author Response
REVIEWER 2
Comment: This file records traces of changes. Please organize the files in an orderly fashion.
Answer: We appreciate the comments on improving the writing on behalf of the authors who contributed to the document.
Best regards
Alejandra Guillermina Miranda-Díaz, MD, PhD
